# Low-Complexity Transmit Power Control for Secure Communications in Wireless-Powered Cognitive Radio Networks

**DOI:** 10.3390/s21237837

**Published:** 2021-11-25

**Authors:** Kisong Lee

**Affiliations:** Department of Information and Communication Engineering, Dongguk University, Seoul 04620, Korea; kslee851105@gmail.com; Tel.: +82-2-2260-3233

**Keywords:** secure communication, energy harvesting, cognitive radio network, secrecy rate, transmit power control

## Abstract

In this study, wireless-powered cognitive radio networks (WPCRNs) are considered, in which *N* sets of transmitters, receivers and energy-harvesting (EH) nodes in secondary networks share the same spectrum with primary users (PUs) and none of the EH nodes is allowed to decode information but can harvest energy from the signals. Given that the EH nodes are untrusted nodes from the point of view of information transfer, the eavesdropping of secret information can occur if they decide to eavesdrop on information instead of harvesting energy from the signals transmitted by secondary users (SUs). For secure communications in WPCRNs, we aim to find the optimal transmit powers of SUs that maximize the average secrecy rate of SUs while maintaining the interference to PUs below an allowable level, while guaranteeing the minimum EH requirement for each EH node. First, we derive an analytical expression for the transmit power via dual decomposition and propose a suboptimal transmit power control algorithm, which is implemented in an iterative manner with low complexity. The simulation results confirm that the proposed scheme outperforms the conventional distributed schemes by more than 10% in terms of the average secrecy rate and outage probability and can also considerably reduce the computation time compared with the optimal scheme.

## 1. Introduction

Cognitive radio networks (CRNs) have received considerable attention as a means to use the unused spectrum to enhance the spectral efficiency (SE) [1]. In CRNs, unlicensed secondary users (SUs) can opportunistically access the spectrum unoccupied by primary users (PUs) as long as they do not interfere with the operation of the PUs [2]. To realize effective spectrum sharing between two networks, several researchers have attempted to mitigate co-channel interference, for instance, through transmit power control (TPC) [3,4], subcarrier allocation [5] and transmit beamforming pattern [6].

In addition to spectrum sharing, there exist many opportunities to harvest energy from radio frequency (RF) signals in CRNs; thus, wireless-powered CRNs (WPCRNs) have actively been investigated [7,8,9,10,11]. The authors of [7,8] proposed a joint information and energy cooperation strategy between PUs and SUs, while cooperative spectrum sensing was optimized under the constraints of collision and energy causality [9]. Moreover, resource allocation strategies have been suggested to maximize the sum throughput of SUs capable of energy harvesting (EH) [10,11].

Notably, the coexistence of different networks is associated with a security issue owing to the increased risk of eavesdropping. Consequently, a number of studies on physical layer security have been performed to guarantee secure communications without relying on a secret key in CRNs [12,13,14,15]. In particular, the security-aware scheduling and power allocation of SUs were optimized to maximize the sum secrecy throughput [12] and resource allocation was proposed to maximize the sum rate of SUs while ensuring the secrecy outage constraint of PUs [13]. In [14], a security-aware proportional fairness resource allocation was developed to improve the sum secrecy throughput and user fairness of SUs. Furthermore, an optimal relay selection was proposed for WPCRNs and the secrecy performances were analyzed with respect to the secrecy outage probability and ergodic rate [15]. Nevertheless, research on WPCRNs remains limited and most of the existing security strategies for CRNs are based on a centralized approach, which involves a high computational complexity [12,13,14]. Distributed TPC strategies were studied in [16,17]; however, these schemes showed not only sub-optimal performances but also could not be directly applied to WPCRNs. Therefore, it is necessary to establish a security-aware TPC algorithm that can be implemented with a low complexity for practical WPCRNs.

Considering this aspect, this research study focuses on secure communications for WPCRNs. The contributions of our work can be summarized as follows:We present a practical system model for WPCRNs, in which multiple SUs share the same spectrum with PUs and EH nodes are not allowed to interpret information but are licensed to collect energy from the transmitted signals.To prevent eavesdropping of untrusted EH nodes and share the spectrum efficiently, an optimization problem is formulated to derive the optimal transmit powers of SU transmitters (Txs) to maximize the average secrecy rate of SUs while ensuring the requirements of allowable interference on PU receiver (Rx) and minimum amount of energy for each EH node. Given that the formulated problem is non-convex, dual decomposition is performed to identify the suboptimal value of the transmit power and develop a low complexity TPC strategy.Performance evaluations based on intensive simulations show that the proposed scheme achieves near-optimal performances in terms of the average secrecy rate and outage probability and significantly reduces the computational complexity compared with the optimal scheme.

The remaining paper is organized as follows: Section 2 presents the system model of the WPCRNs, along with the problem statement. Section 3 describes the low-complexity TPC algorithm. Section 4 describes the performance evaluation of the proposed scheme based on extensive simulations. The concluding remarks are presented in Section 5.

## 2. System Model and Problem Statement

Figure 1 shows the system model of WPCRNs for secure communications, in which all nodes are equipped with a single antenna. In particular, *N* SU Tx–Rx pairs share the same frequency with PUs as long as the amount of interference on PUs is less than an allowable level. Moreover, *N* EH nodes associated with each SU Tx–Rx pair are not permitted to decode information and are only allowed to harvest energy from the transmitted signals. Consequently, these nodes are regarded as untrusted nodes [18]. The sets of SU Tx–Rx pairs and their associated EH nodes are denoted as N, i.e., |N|=N. The wireless channel between SU Tx *i* and SU Rx *j* is denoted as $h_{i,j}$ and that between SU Tx *i* and EH node *j* is denoted as $g_{i,j}$. In addition, the index 0 corresponds to PUs, i.e., $h_{i,0}$ is the wireless channel between SU Tx *i* and PU Rx and $g_{0,i}$ is the wireless channel between PU Tx and EH node *i*. It is assumed that all wireless channels follow a discrete time block-fading model and $h_{i,0}$ is available at SU Tx *i* to maintain the amount of interference on PU Rx below an acceptable level [19].

The signal received at SU Rx *i* is expressed as
(1)$$\begin{array}{c} y_{i}=\sqrt{p_{i}}h_{i,i}x_{i}+\sum_{k\in N\backslash \{i\}}\sqrt{p_{k}}h_{k,i}x_{k}+\sqrt{p_{0}}h_{0,i}x_{0}+z_{i}, \end{array}$$
where $x_{i}$ and $x_{0}$ indicate the normalized data sent by SU Tx *i* and PU Tx with transmit power $p_{i}$ and $p_{0}$, respectively; $z_{i}∼\mathrm{CN}\left(0,\sigma^{2}\right)$ represents additive white Gaussian noise.

Using (Equation 1), the achievable SE can be specified as
(2)$$\begin{array}{cc} r_{i} & =\log_{2}\left(1+\frac{p_{i}{\left|h_{i,i}\right|}^{2}}{\sum_{k\in N\backslash \{i\}}p_{k}|h_{k,i}{|}^{2}+p_{0}{\left|h_{0,i}\right|}^{2}+\sigma^{2}}\right). \end{array}$$

The interference caused by SU Tx *i* on PU Rx is expressed as
(3)$$\begin{array}{cc} I_{i} & =p_{i}{\left|h_{i,0}\right|}^{2}. \end{array}$$

Moreover, the signal received at EH node *i* is obtained as
(4)$$\begin{array}{c} {\hat{y}}_{i}=\sqrt{p_{i}}g_{i,i}x_{i}+\sum_{k\in N\backslash \{i\}}\sqrt{p_{k}}g_{k,i}x_{k}+\sqrt{p_{0}}g_{0,i}x_{0}+{\hat{z}}_{i}, \end{array}$$
where ${\hat{z}}_{i}∼\mathrm{CN}\left(0,\sigma^{2}\right)$. Because each EH node can collect energy from the signals sent by all Txs, the total energy harvested at EH node *i* is represented by
(5)$$\begin{array}{cc} e_{i} & =\zeta_{i}\left(p_{0}|g_{0,i}{|}^{2}+\sum_{j\in N}p_{j}{\left|g_{j,i}\right|}^{2}\right), \end{array}$$
where $\zeta_{i}$ is the energy conversion efficiency at EH node *i*. If EH node *i* stops harvesting energy and decides to eavesdrop on information from SU Tx *i*, its achievable SE is defined as
(6)$$\begin{array}{cc} r_{i}^{e} & =\log_{2}\left(1+\frac{p_{i}{\left|g_{i,i}\right|}^{2}}{\sum_{k\in N\backslash \{i\}}p_{k}|g_{k,i}{|}^{2}+p_{0}{\left|g_{0,i}\right|}^{2}+\sigma^{2}}\right). \end{array}$$

Using (Equation 2) and (Equation 6), the secrecy rate of set *i* can be defined as follows [20]:(7)$$\begin{array}{cc} r_{i}^{s} & ={\left[r_{i}-r_{i}^{e}\right]}^{+}, \end{array}$$
where ${\left[\cdot \right]}^{+}=\max\left(0,\cdot \right)$.

To ensure the information secrecy from untrusted EH nodes while supplying a sufficient amount of energy to each EH node, $E_{\min}$, and maintaining the interference to PU Rx below an allowable level, $I_{\max}$, the transmit powers of SU Txs must be optimized. Accordingly, the problem of average secrecy rate maximization can be formulated as follows:(8)$$\begin{array}{cc} \max_{0⪯\vec{p}} & \;\;\;\;\frac{1}{N}\sum_{i\in N}r_{i}^{s}\\ s.t. & \;\;\;\;\sum_{i\in N}I_{i}\leq I_{\max}\\ \; & \;\;\;\;e_{i}\geq E_{\min},\;\;i\in N\\ \; & \;\;\;\;p_{i}\leq P_{\max},\;\;i\in N, \end{array}$$
where $\vec{p}=\left\{p_{1},p_{2},\cdots ,p_{N}\right\}$ and $P_{\max}$ denotes the maximum transmit power for SU Txs. The co-channel interference of the objective function renders the problem (Equation 8) non-convex; thus, it is difficult to derive the optimal value of $\vec{p}$ in an analytical form. The near-optimal solution can be numerically determined through an exhaustive search (ES), in which all possible combinations are examined by quantizing $\vec{p}$ into *Q* equally spaced values. However, this process requires the complete channel state information (CSI) and extremely high computational complexity of $O\;\left(Q^{N}\right)$; thus, the process has an exponential complexity with respect to *N*.

## 3. Low-Complexity Transmit Power Control Algorithm

In this section, dual decomposition is used to find a suboptimal TPC algorithm with low complexity.

First, the original problem (Equation 8) is decomposed into *N* subproblems and each subproblem is simultaneously solved. In each subproblem, the objective is changed to identify the transmit power of SU Tx that maximizes its own secrecy rate while ensuring the constraints of $I_{\max}$ and $E_{\min}$, which is formulated as follows:(9)$$\begin{array}{cc} \max_{0\leq p_{i}} & \;\;\;\;r_{i}^{s}\\ s.t. & \;\;\;\;I_{i}\leq \frac{I_{\max}}{N}\\ \; & \;\;\;\;e_{i}\geq E_{\min}\\ \; & \;\;\;\;p_{i}\leq P_{\max}. \end{array}$$

In (Equation 9), each SU Tx regulates the interference to PU Rx smaller than $\frac{I_{\max}}{N}$ to meet the constraint, $\sum_{i\in N}I_{i}\leq I_{\max}$.

To solve problem (Equation 9), we develop the Lagrangian function of (Equation 9) as follows [21]:(10)$$\begin{array}{cc} L(p_{i},\lambda_{i},\mu_{i},\kappa_{i})\; & =\;r_{i}^{s}+\lambda_{i}\;\left(\frac{I_{\max}}{N}\;-\;I_{i}\right)+\mu_{i}\;\left(e_{i}\;-\;E_{\min}\right)+\kappa_{i}\;\left(P_{\max}\;-\;p_{i}\right), \end{array}$$
where $\lambda_{i}\geq 0$, $\mu_{i}\geq 0$ and $\kappa_{i}\geq 0$ are the Lagrange multipliers of each constraint in (Equation 9).

The dual problem of (Equation 9) can be formulated as follows [22]:(11)$$\begin{array}{c} \min_{0\leq \lambda_{i},\;0\leq \mu_{i},\;0\leq \kappa_{i}}\;\;F\left(\lambda_{i},\mu_{i},\kappa_{i}\right), \end{array}$$
where $F\left(\lambda_{i},\mu_{i},\kappa_{i}\right)=\max_{p_{i}\geq 0}\;\;L\left(p_{i},\lambda_{i},\mu_{i},\kappa_{i}\right)$. In this framework, the transmit power is updated to maximize $L(p_{i},\lambda_{i},\mu_{i},\kappa_{i})$, whereas the Lagrange multipliers are updated to minimize $F(\lambda_{i},\mu_{i},\kappa_{i})$ in each SU Tx in an iterative manner.

To derive the suboptimal value of $p_{i}$, the Karush–Kuhn–Tucker (KKT) conditions [21] are established with complementary slackness, as follows: (12)$$\begin{array}{cc} \; & \frac{\partial L(p_{i},\lambda_{i},\mu_{i},\kappa_{i})}{\partial p_{i}}\;=\;\frac{\partial r_{i}^{s}}{\partial p_{i}}\;-\;\lambda_{i}\frac{\partial I_{i}}{\partial p_{i}}\;+\;\mu_{i}\frac{\partial e_{i}}{\partial p_{i}}\;-\;\kappa_{i}\;=\;0, \end{array}$$(13)$$\begin{array}{cc} \; & \lambda_{i}\left(\frac{I_{\max}}{N}-I_{i}\right)=0, \end{array}$$(14)$$\begin{array}{cc} \; & \mu_{i}\left(e_{i}-E_{\min}\right)=0, \end{array}$$(15)$$\begin{array}{cc} \; & \kappa_{i}\left(P_{\max}-p_{i}\right)=0, \end{array}$$(16)$$\begin{array}{cc} \; & 0\leq p_{i},\;\;0\leq \lambda_{i},\;\;0\leq \mu_{i},\;\;0\leq \kappa_{i}. \end{array}$$

The transmit power that satisfies the KKT conditions in (Equation 12)–(16) can be derived as
(17)$$\begin{array}{c} p_{i}\;=\;{\left[\frac{1}{\ln2\;\left(\lambda_{i}|h_{i,0}{|}^{2}\;-\;\mu_{i}\zeta_{i}{\left|g_{i,i}\right|}^{2}\;+\;\kappa_{i}\right)\;+\;t_{i}^{\left[s\right]}}\;-\;\frac{\Psi_{i}}{|h_{i,i}{|}^{2}}\right]}^{\;+}\;\;, \end{array}$$
where $\Psi_{i}=\sum_{j\in N\backslash \{i\}}p_{j}|h_{j,i}{|}^{2}\;+\;p_{0}{\left|h_{0,i}\right|}^{2}\;+\;\sigma^{2}$ and $t_{i}^{\left[s\right]}$ is defined as
(18)$$\begin{array}{c} t_{i}^{\left[s\right]}=\frac{|g_{i,i}{|}^{2}}{R_{i}}, \end{array}$$
where $R_{i}=\sum_{l\in N}p_{l}|g_{l,i}{|}^{2}+p_{0}{\left|g_{0,i}\right|}^{2}+\sigma^{2}$. In (Equation 17), $\Psi_{i}$ can be easily measured at SU Rx *i* by subtracting the signal power from SU Tx *i* from the total received power. Moreover, $t_{i}^{\left[s\right]}$ can be calculated at EH node *i* because the denominator of (Equation 18) is the total received power. Therefore, the transmit power of SU Tx *i* can be determined as shown in (Equation 17) by sharing information within set *i* without other sets j∈N\{i}, which allows the proposed algorithm to operate with low complexity.

Each Lagrange multiplier is updated using a gradient algorithm as follows [22]:(19)$$\begin{array}{cc} \lambda_{i} & \leftarrow {\left[\lambda_{i}-\nu_{1}\left(\frac{I_{\max}}{N}-I_{i}\right)\right]}^{+},\\ \mu_{i} & \leftarrow {\left[\mu_{i}-\nu_{2}\left(e_{i}-E_{\min}\right)\right]}^{+},\\ \kappa_{i} & \leftarrow {\left[\kappa_{i}-\nu_{3}\left(P_{\max}-p_{i}\right)\right]}^{+}, \end{array}$$
where $\nu_{1}$, $\nu_{2}$ and $\nu_{3}$ are step sizes to update each Lagrange multiplier.

The proposed algorithm operates as described in Algorithm 1. Because $\epsilon^{-2}$ iterations are required to ensure that the norm of the gradient is smaller than ϵ in the worst case [23], the computational complexity of the proposed algorithm is $O\;\left(N\epsilon^{-2}\right)$, where *N* is the number of computations for the calculation of $\vec{p}$. The proposed algorithm exhibits a linear complexity with respect to *N*, which corresponds to a significant reduction in the computational complexity compared to that of the ES. It is worth noting that the centralized approach hinders real-time operability as it requires a long computation time to determine the TPC strategy, but the low complexity of the proposed scheme improves its practical applicability for WPCRNs.
**Algorithm 1** Low-complexity transmit power control algorithm1: Randomly initialize $\vec{p}$, $\vec{\lambda }$, $\vec{\mu }$ and $\vec{\kappa }$2: **repeat**3:  ${\vec{p}}_{\mathrm{old}}\leftarrow \vec{p}$4:  **for**i=1 to *N*5:   Compute $t_{i}^{\left[s\right]}$ according to (Equation 18)6:   Compute $p_{i}$ according to (Equation 17)7:   Update $\lambda_{i}$, $\mu_{i}$ and $\kappa_{i}$ according to (Equation 19)8:  **end for**9:  $\vec{p}=\left\{p_{1},p_{2},\cdots ,p_{N}\right\}$10: **until**$∥\vec{p}-{\vec{p}}_{\mathrm{old}}∥<\epsilon$

## 4. Simulation Results and Discussion

To evaluate the performance of the proposed scheme, the following system parameters, shown in Table 1, were set as default: *N* = 3, $P_{\max}$ = $p_{0}$ = 30 dBm, $\sigma^{2}=-100$ dBm, $E_{\min}=-15$ dBm, $I_{\max}=-50$ dBm and $\zeta_{i}$ = 0.5 for i∈N [19,24]. Considering device-to-device communications underlay cellular networks as the target system of WPCRNs [16], the SUs and EH nodes were generated randomly over an area of 50 m × 50 m, with the maximum distances of each signal link and EH link in the same node set defined as 15 m and 10 m, respectively. Moreover, the secondary network was located, on average 1 km, from PUs for reliable spectrum sharing. To generate the wireless channels, a simplified path loss model with a path loss exponent of 2.7 was considered for urban areas [18]. In addition, for multi-path fading, Rayleigh fading was used for the signal links to reflect the characteristics of non-line-of-sight (nLoS), while Rician fading with a *K*-factor of six was used for the EH links to reflect the characteristics of line-of-sight (LoS) [24]. The performance metrics are the average secrecy rate and outage probability, which can be mathematically expressed as $E\left[\left(1_{\sum_{i\in N}I_{i}\leq I_{\max}}\right)\cdot \left(\prod_{i\in N}1_{e_{i}\geq E_{\min}}\right)\cdot \frac{1}{N}\sum_{i\in N}r_{i}^{s}\right]$ and $E\left[1-\left(1_{\sum_{i\in N}I_{i}\leq I_{\max}}\right)\cdot \left(\prod_{i\in N}1_{e_{i}\geq E_{\min}}\right)\right]$, respectively. The average secrecy rate was set as zero if the constraint of $I_{\max}$ or $E_{\min}$ was violated to impose a penalty. We also considered the following five schemes for performance evaluation:Optimal scheme: With the complete CSI at SU Txs, the ES can attain near-optimal performance with Q=100.Proposed TPC scheme: The transmit powers of SU Txs are determined using the proposed algorithm described in Algorithm 1.Barrier scheme: The transmit powers of SU Txs are determined by a centralized barrier method [21,25], which is one of interior-point methods.Binary TPC scheme: The transmit power of each SU Tx is determined as either $P_{\max}$ or zero in the direction of maximizing the average secrecy rate [17].Maximum power scheme: The transmit powers of SU Txs are always determined as their maximum transmit powers, $P_{\max}$.Random power scheme: The transmit powers of SU Txs are randomly determined.

Figure 2 shows the average secrecy rate and outage probability against the allowable interference level ($I_{\max}$). As $I_{\max}$ decreased, it became more difficult to satisfy the constraint of $I_{\max}$. Therefore, for conventional distributed schemes, e.g., binary TPC, maximum power and random power schemes, the outage probability increased and the average secrecy rate significantly degraded. Specifically, although the binary scheme took values of either $P_{\max}$ or zero for the transmit power, it implemented adaptive TPC and exhibited the highest performance among the conventional distributed schemes. The maximum power scheme outperformed the random power scheme, but the performance trend reverses as $I_{\max}$ decreased due to a critical violation of the constraint of $I_{\max}$. In contrast, the proposed scheme effectively regulated the amount of interference to PU Rx to the allowable level by the effective TPC even when $I_{\max}=-60$ dBm. Therefore, the performance is comparable to that of the optimal scheme and centralized barrier scheme and the proposed scheme outperformed the conventional distributed schemes for the entire range of $I_{\max}$.

Figure 3 shows the average secrecy rate and outage probability against the required harvested energy ($E_{\min}$). As in the case shown in Figure 2, it was hard to provide adequate energy to EH nodes to satisfy the EH requirements as $E_{\min}$ increased. Consequently, the average secrecy rate decreased and the outage probability increased for all considered schemes when $E_{\min}>-10$ dBm. In particular, the performance of the binary and random schemes was inferior to that of the proposed scheme because these schemes could not adaptively perform TPC according to variations in the channel realization or constraints. The performance of the maximum power scheme was relatively constant against the change in $E_{\min}$ because it always uses the maximum transmit power; however, the scheme achieved the worst performance due to excessive interference among node sets. In contrast, the proposed scheme exhibited near-optimal performance for the entire range of $E_{\min}$.

Figure 4 shows the average secrecy rate, outage probability and computation time against the number of node sets (*N*). As *N* increased, the node sets exhibited severe interference with one another. Therefore, the average secrecy rate decreased with *N* for all the considered schemes. The maximum and random power schemes corresponded to the least computation time because no computation was required to determine the transmit powers in these schemes and computation time for the binary scheme was less than that for the proposed scheme. Although the proposed scheme required a relatively high computation time to determine the transmit powers, it outperformed the conventional distributed schemes in terms of the average secrecy rate and outage probability owing to the implementation of proper TPC to reduce interference. Moreover, the performance of the proposed scheme is comparable to that of the optimal scheme with a significantly reduced computation time. The centralized barrier scheme performed slightly better than the proposed method, but required high computation time and knowledge of perfect CSI. This result demonstrates the effectiveness of the proposed scheme in achieving secure communications with a reasonable computational complexity.

Figure 5 shows the cumulative distribution function (CDF) versus the average secrecy rate. The CDF of the proposed scheme was the closest to that of the optimal scheme, which demonstrates the near-optimality of the proposed TPC strategy. Moreover, higher average secrecy rates were noted in the CDFs of the optimal and proposed schemes than in the existing distributed schemes, which indicates that the adaptive TPC is crucial to enhance the average secrecy rate.

## 5. Conclusions

A low-complexity TPC algorithm to realize secure communications in WPCRNs was developed, in which the transmit powers of SU Txs are optimized to maximize the average secrecy rate while ensuring the requirements of allowable interference on PUs and the minimum amount of energy for each EH node. In particular, the closed-form equation for the transmit power of each SU Tx was mathematically derived via dual decomposition and a low-complexity TPC algorithm implemented in an iterative manner was established. Simulations performed under various scenarios highlight that the proposed scheme could enhance the average secrecy rate and outage probability, e.g., at least 10% or more compared to the existing schemes at default settings, owing to the use of the adaptive TPC strategy. Moreover, the proposed scheme reduced the computation time compared to the optimal scheme, such that its computation time was less than tens of milliseconds even when the number of nodes was large.

## Figures and Tables

**Figure 1 sensors-21-07837-f001:**
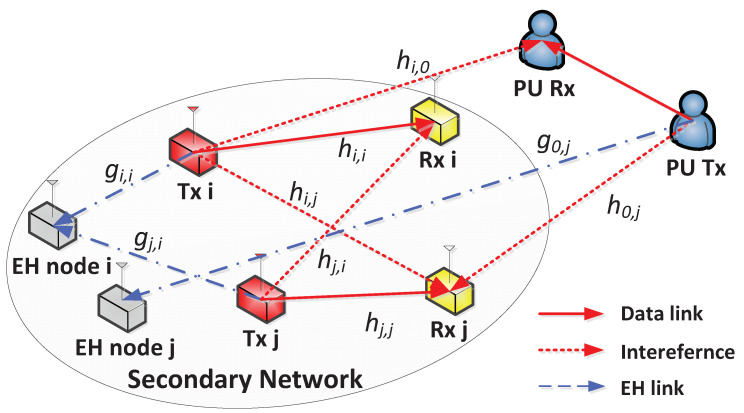
Example of WPCRNs for secure communications with N=2.

**Figure 2 sensors-21-07837-f002:**
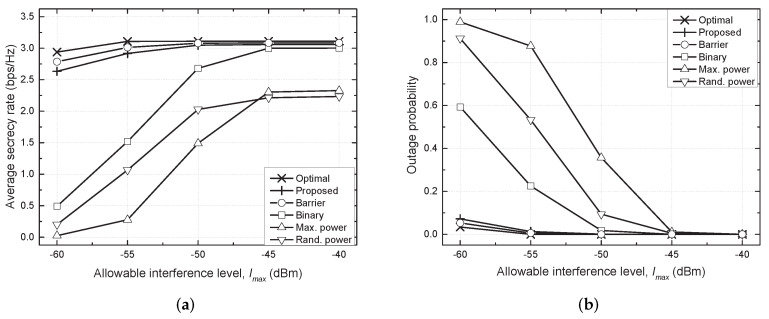
Performance comparison against allowable interference level ($I_{\max}$). (**a**) Average secrecy rate vs. $I_{\max}$. (**b**) Outage probability vs. $I_{\max}$.

**Figure 3 sensors-21-07837-f003:**
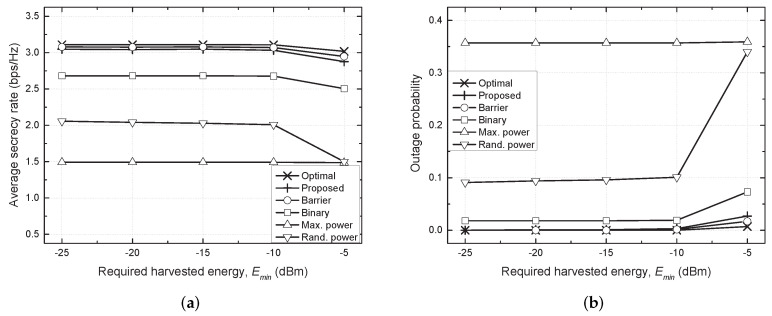
Performance comparison against required harvested energy ($E_{\min}$). (**a**) Average secrecy rate vs. $E_{\min}$. (**b**) Outage probability vs. $E_{\min}$.

**Figure 4 sensors-21-07837-f004:**
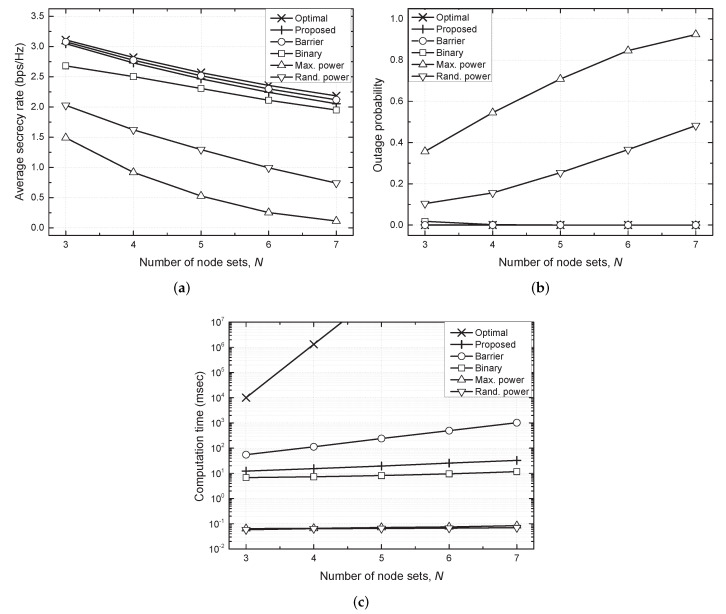
Performance comparison against number of node sets (*N*). (**a**) Average secrecy rate vs. *N*. (**b**) Outage probability vs. *N*. (**c**) Computation time vs. *N*.

**Figure 5 sensors-21-07837-f005:**
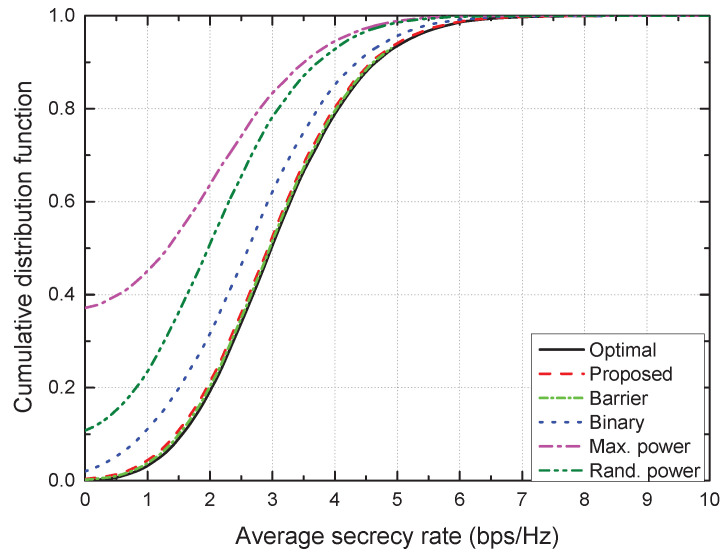
CDF vs. average secrecy rate.

**Table 1 sensors-21-07837-t001:** Parameter Setup.

Parameter	Value
Number of node sets	*N* = 3
Maximum transmit power for SU Txs	$P_{\max}$ = 30 dBm
Transmit power for PU Tx	$p_{0}$ = 30 dBm
Noise power	$\sigma^{2}=-100$ dBm
Required harvested energy	$E_{\min}=-15$ dBm
Allowable interference level	$I_{\max}=-50$ dBm
Energy conversion efficiency for EH nodes	$\zeta_{i}$ = 0.5 for i∈N
Size of area for distributing nodes	50 m × 50 m
Path loss exponent	2.7
*K*-factor of Rician fading	6

## Data Availability

Not applicable.

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
