# Peer review of "Low-Complexity Transmit Power Control for Secure Communications in Wireless-Powered Cognitive Radio Networks"

_sensors, 2021, doi:10.3390/s21237837_

Round 1

Reviewer 1 Report

The authors address an interesting and relevant topic of research, i.e. spectral efficiency has received considerable attention from academics. The authors aim at optimizing the communications between wireless nodes, namely the performance metrics secrecy rate, outage probability and algorithm complexity by reducing interference between them through the power transmission optimization. The reduction in complexity of such solutions is nuclear for the correct operation of these solutions, given that the wireless environment is quite volatile/dynamic, thus computing the optimal power transmission must be fast to ensure that it is the optimal value for the current wireless channel conditions. I have some comments that if considered might improve the overall quality of this document.

In the abstract, the author does not start by identifying the problem that they want to address, there is a lack of contextualization, the same occurs in the Introduction section. The abstract set off with a description of the architecture of the wireless environment. It might be relevant for the reader, considering that in this field of study there are several topics, research works focus only on the power transmission, or simply in the energy harvesting, etc. Here, all these features are included in a single algorithm.

The Introduction has no contextualization, and the problem and the importance of such solutions are not described. The authors included a satisfactory number of related works, however, I could not find in this set of solutions any solution having a low level of computational complexity. Moreover, I invite the authors to include in this section the contribution of their work.

Please explain to the readers why a centralized approach is considered a high computational complexity solution and how it negatively affects the effectiveness of these solutions. I do believe that the authors when referring to complexity want to emphasize the processing time is high and thus solutions fail to apply the optimal value at the exact moment that data is broadcasted.

It seems to meet that most of these formulas are often used in the do analytical describe the wireless channels, if that is the case please add the respective references.

The proposed solution shows to be quite efficient for most of the metrics here considered, however, the processing time is high, and, as the authors simply compared their solution against conventional solutions with few or no processing required, the proposed solution processing time is drastically higher than results of the remaining solutions not providing to the reader the expected message: this is a low complexity solution. I invite the authors to include a high computational complexity solution identified in the related work, to effectively show the improvements achieved in this metric.

During the performance analysis in section 4, or at least in the text of the conclusion section, consider comparing the performance as a percentage to give a better idea in which magnitude the proposed solution overcome the remaining solutions.  

Author Response

First of all, we would like to sincerely thank the reviewer for valuable comments. We have considered all the reviewer's comments and suggestions and have modified our manuscript accordingly. We have also prepared detailed responses to each of the concerns raised. Once again, we would like to thank you for your careful corrections and suggestions.

Sincerely,

Kisong Lee

Reviewer 2 Report

  1. In the abstract, the author mentioned comparison is made with the optimal scheme. However, in the methodology, it is discovered that comparison is made with optimal, several other sub-optimal schemes as well - please revise the abstract.
  2. Please use italic to represent vector and bold to represent a matrix in the equations and drawing, especially Figure 1. 
  3. In the first paragraph of Section 4, please present relevant system parameters in a table format and include the definition of each of the parameters in a separate column to simplify reader understanding. 
  4. In Line 100, Page 5, the considered simulation area is very small. It is suggested that the author relate with potential WPRCN application that can be implemented under such an area. 

Author Response

(The authors gave the same response as above.)
